# Post-COVID-19 Vaccination Myocarditis: A Histopathologic Study on a Monocentric Series of Six Cases

**DOI:** 10.3390/diagnostics14020219

**Published:** 2024-01-19

**Authors:** Hyo-Suk Ahn, Yuran Ahn, Jaehyuk Jang, Seonghyun Bu, Sungmin Lim, Chanjoon Kim, Jong-Min Lee, Kyungji Lee, Kyung-Jin Seo

**Affiliations:** 1Division of Cardiology, Department of Internal Medicine, Uijeongbu St. Mary’s Hospital, The Catholic University of Korea, Seoul 06591, Republic of Korea; alaco0502@gmail.com (H.-S.A.);; 2Catholic Research Institute for Intractable Cardiovascular Disease (CRID), College of Medicine, The Catholic University of Korea, Seoul 06591, Republic of Korea; 3Midwest International Research Institute, Midwest University, 851 Parr Rd., Wentzville, MO 63385, USA; 4Department of Hospital Pathology, College of Medicine, The Catholic University of Korea, Seoul 06591, Republic of Korea

**Keywords:** myocarditis, COVID-19 vaccination, endomyocardial biopsy

## Abstract

Many reports on the development of myocarditis following coronavirus disease 2019 (COVID-19) vaccination (PCVM) have emerged. However, only a few case studies have investigated endomyocardial biopsy (EMB) results. This study describes the clinicopathologic features of PCVM. We surveyed all hospitalized patients in a single university hospital in Korea and identified six cases of PCVM. All six patients underwent EMB, five of whom were men aged 15–85 years. All patients developed cardiac dysfunction. Among these patients, two had mild disease without sequelae, whereas the other four had dilated cardiomyopathy with depressed cardiac function. All six cases demonstrated lymphohistiocytic myocarditis. Two of our cases fulfilled the criterion of CD3+ T lymphocytes > 7 cells/mm^2^ (Case nos. 3 and 6), while the remaining four cases did not fulfill the Dallas criteria. In conclusion, most PCVM cases showed mild degree inflammation histopathologically, and some cases could not fulfill the Dallas criteria and were classified as borderline myocarditis.

## 1. Introduction

Myocarditis has various infectious and noninfectious etiologies, including viruses; autoimmune diseases, such as sarcoidosis; immune stimulation, such as from vaccines or cancer therapies; and exposure to toxins and drugs, including endogenous biochemical compounds, as seen in amyloidosis and in thyrotoxicosis [1]. 

Myocarditis has been recognized as a rare complication of the coronavirus disease 2019 (COVID-19) mRNA vaccinations, particularly in male adolescents and young adults [2]. According to a review summarizing 61 case reports or series on post-COVID-19 vaccination myocarditis (PCVM), the mean age of patients was 26 years old (range: 14–56), and 98% were male [2,3]. However, most of these studies reported no histopathological data obtained by endomyocardial biopsy (EMB) [2,4,5,6], and only a few case studies have reported EMB results (Table 1) [7,8,9,10,11,12]. 

The histopathologic spectrum of myocarditis is very broad, and myocarditis can be categorized according to the prevalent histopathologic pattern including lymphocytic, lymphohistiocytic, eosinophilic, and neutrophilic myocarditis and giant-cell myocarditis, as well as myocarditis with granulomata [13]. Chow et al. described a case of lymphohistiocytic myocarditis in which the histopathological findings of the EMB specimen showed patchy endocardial and intra-myocardial lymphohistiocytic infiltrates with scattered eosinophils and focal myocyte injury [7]. Immunohistochemical CD3 and CD68 staining confirmed the lymphocytic and histiocytic nature of the infiltrate, respectively. An ill-defined granuloma was also observed. The authors argued that these findings were suggestive of a post-vaccination hypersensitivity reaction. Kiblboeck et al. recently reported a case series of three young male patients with PCVM following vaccination with the BNT162b2 mRNA COVID-19 vaccine from BioNTech/Pfizer, including detailed histopathologic features of EMB [8].

In this case series, we describe six additional cases of suspected PCVM with detailed EMB histopathologic findings.

## 2. Materials and Methods

### Case Identification and Clinical Data

Following our index case (Case no. 1 in Table 1), we performed an archival search using a free-text search tool in our pathology database to identify cases diagnosed by EMB at our institution from August 2021 to March 2022. We first looked for patients who required hospitalization among patients who clinically showed symptoms of myocarditis, such as sudden chest pain, fever, and heart failure, at our institution from August 2021 to March 2022. Among these patients, we re-selected those who had received a coronavirus vaccination 3 weeks prior to recent hospitalization. Among these patients, we excluded cases with the possibility of other causes, such as viral or autoimmune diseases: exclusion criteria are positive serological marker tests for other viruses, such as adenovirus, coxsackievirus B1, and parvovirus, or any evidence of autologous immune disease. Among these, patients who underwent EMB were selected, and six cases were identified. Coronary computed tomographic angiography or coronary angiography showed normal coronary arteries with no stenosis or occlusion in these six cases. All available diagnostic slides were retrieved from the pathology department archives and reviewed by one pathologist (KJ Seo). Clinical information and follow-up data were obtained from electronic medical records.

## 3. Results

The clinicopathologic features of previous PCVM cases and our six cases are summarized in Table 1.

### 3.1. Clinicopathologic Features

Six EMB cases were identified and reviewed. A summary of the clinical features is presented in Table 1. Briefly, patients were aged 19–83 years old (median age 55.5 years). Four patients were vaccinated with the BioNTech/Pfizer BNT162b2 vaccine, whereas two were vaccinated with the Moderna mRNA-1273 vaccine before EMB. Cardiac symptoms developed 2–19 days after inoculation with the first, second, or third COVID-19 vaccine dose. The severity of cardiac dysfunction varied; left ventricular ejection fraction (LVEF) ranged from 18% to 65% (Table 1). 

The symptoms of two patients were mild, with a normal range of LVEF, and had no sequelae (Case no. 3, LVEF 65%; Case no. 5, 60%; Table 1). However, these patients showed increased levels of myocardial enzymes, such as troponin-T and creatine kinase MB isoenzyme (CK-MB), and some delayed enhancement in the left ventricle on cardiac magnetic resonance imaging (MRI) (Figure 1). The T2-weighted black blood image (left) and delayed enhancement image (right) of Case no. 1 showed diffuse increased signal intensity (as compared with skeletal muscles) due to edema and abnormal subepicardial and mid-wall hyper-enhancement, particularly in the anterior and antero-septal portions of the left ventricle (in part and modified from Figure 1 in [14]). The other four patients (Case nos. 1, 2, 4, and 6) exhibited depressed LVEF with dilated cardiomyopathy (DCM). Furthermore, three of these four showed some delayed enhancement in the left ventricle on heart MRI. The assessment could not be performed on the fourth patient (Case no. 2) because of hypoxic brain damage caused by ventricular fibrillation (VF). A cardiac MRI could not be performed because breathing was not controlled, and myocardial enzymes (e.g., troponin-T and CK-MB) were elevated. Another patient (Case no. 4) underwent surgery for an implantable cardioverter-defibrillator for recurrent ventricular tachycardia. Overall, two patients (Case nos. 3 and 5) with mild symptoms received conservative therapy, whereas the remaining four (Case nos. 1, 2, 4, and 6) received treatment for heart failure. No immunosuppressive treatments were administered.

The quantitative analysis of infiltrating CD3+ lymphocytes and CD68+ histiocytes of reported PCVM cases and our six cases are summarized in Table 2.

### 3.2. Quantitative Analysis of Immunohistochemical Stains

All six cases showed various degrees of myocarditis with a few microfoci of myofiber eosinophilic changes and disrupted myocytes and inflammatory infiltrates (Figure 2, Figure 3 and Figure 4). Two cases met the >7 CD3-positive T lymphocytes/mm^2^ Dallas criterion (Case nos. 3 and 6), but the remaining four did not. However, three cases met the ≥14 leukocytes in the myocardium criterion, including up to 4 monocytes/mm^2^ (Case nos. 1, 2, 3, and 6), whereas the remaining three did not. These (Case nos. 2, 4, and 5) could be categorized as “borderline” myocarditis, and repeat biopsy may be indicated (Table 2).

In Cases 1–6, the histopathologic type of myocarditis was lymphohistiocytic myocarditis, showing 2–10 CD3-positive infiltrating lymphocytes/mm^2^ and 0–10 CD68-positive infiltrating histiocytes/mm^2^ (Figure 2, Figure 3 and Figure 4, Table 2). In Case 4, the histopathological type was lymphocytic myocarditis (LM), showing 1 CD3-positive lymphocytes/mm^2^ and 7 CD68-positive histiocytes/mm^2^ were identified and classified as borderline myocarditis (Figure 4, Table 2). 

## 4. Discussion

Many heterogeneous etiologies can cause inflammatory cardiomyopathy, which is characterized by inflammatory cell infiltration into the myocardium and a high risk of deteriorating cardiac function [1]. Various factors, including viral, bacterial, protozoal, or fungal infections, as well as various toxic substances or drugs and systemic immune-mediated diseases, can cause inflammatory cardiomyopathy.

The clinical presentation of PCVM may include reduced LVEF, heart failure, advanced atrioventricular block, sustained ventricular arrhythmias, and, in severe cases, cardiogenic shock with an increased risk of death or need for heart transplantation. However, most cases of PCVM are mild [1]. According to one study that evaluated 238 patients with PCVM, most were male (87.1%), and the mean ± standard deviation age was 27.4 ± 16 (range 12–80) years [15]. The most common symptom was chest pain (93%). Approximately 30% of the patients had reduced LVEF, but more than half recovered on repeat imaging. Of the 238 patients, 11 developed cardiogenic shock, and 5 patients (1.7%) died. Chow et al. [7] pointed out some intriguing distinctions between the demographics and clinical history of one reported patient compared with previously documented cases of severe acute respiratory syndrome coronavirus 2 vaccination myocarditis in a case report.

In the present study, all six patients underwent EMB. The symptoms of the two patients were mild, with a normal range of left ventricular function, and had no sequelae. Two patients with mild symptoms received conservative therapy, whereas the remaining four received treatment for heart failure. No immunosuppressive treatments were administered.

Currently, most cardiac diseases can be diagnosed by non-invasive procedures such as echocardiography and cardiac MRI. Despite some limitations, such as inconsistencies in diagnosis, a group of pathological conditions that require biopsy for a conclusive diagnosis remains. These include myocarditis, amyloidosis, sarcoidosis, and giant-cell myocarditis [16]. Khan et al. [16] reported that in 29.2% of cases (73 out of 250), the results of EMB histology significantly influenced and altered patient management. Although the diagnostic utility of EMB for myocarditis is limited, the authors suggest that minimal lymphohistiocytic infiltration should raise suspicion for myocarditis. Histological diagnosis of myocarditis in EMB specimens is still based on the Dallas criteria. These criteria account for the distribution, extent, and cell types of inflammatory infiltrates [13,17,18]. To reduce inter-observer variability and establish a cut-off for defining abnormal lymphohistiocytic infiltration, the European Society of Cardiology introduced quantitative immunohistochemistry (IHC) criteria for diagnosing myocarditis: ≥14 leukocytes in the myocardium, including up to 4 monocytes/mm^2^, with the presence of >7 CD3-positive T lymphocytes/mm^2^. These IHC criteria may reduce inter-observer variations and thus enhance the diagnostic value of EMB for suspected myocarditis. In our study, three cases met the Dallas criteria, whereas the remaining three did not and were categorized as “borderline” myocarditis.

Of the eight cases described in the Larson et al. [5] case series, EMB was conducted with only one patient who did not demonstrate myocardial infiltrate before steroid initiation. In the report by Chow et al., EMB findings revealed lymphohistiocytic-type myocarditis. In a systematic review of the cumulative experience of 20,212 cases of PCVM that identified 238 positive patients, only 13 underwent an EMB or autopsy. Furthermore, 10/13 (77%) patients showed abnormal histopathology suggestive of myocarditis [15]. In our hospital’s experience, myocarditis due to other causes has a much more severe inflammatory response pathologically; however, the disease severity of PCVM is relatively mild. Although not pathologically severe, PCVM cases were clinically severe with conditions such as DCM, VF, and ventricular tachycardia. Therefore, histopathological confirmation of myocarditis is important in predicting prognosis. 

In terms of the molecular pathogenesis of PCVM, we conducted single-cell RNA sequencing and single-cell T-cell receptor sequencing analyses of peripheral blood mononuclear cells (PBMCs) obtained from one of the patients (Case no. 1) and published the result implicating the immunologic background of PCVM: the greatest changes were observed in the transcriptomic profile of monocytes in terms of the number of differentially expressed genes [14]. 

## 5. Conclusions

Our report describes a series of clinically diagnosed PCVM cases and highlights the potential significance of EMB and histopathological findings in the diagnosis of PCVM. In our cases, the degree of inflammatory cell infiltration was mild, and 50% (3/6) did not fulfill the Dallas criteria. Therefore, these cases could be categorized as “borderline” myocarditis. Interestingly, inflammatory cell infiltration was not proportional to the severity of clinical symptoms. 

Myocarditis is a rare complication of COVID-19 vaccinations, predominantly in male adolescents and young adults. The symptoms in most suspected cases of PCVM are mild; thus, EMB was often not performed. Most PCVM cases showed a mild degree of inflammation histopathologically, and some cases could not fulfill the Dallas criteria and were classified as borderline myocarditis.

## Figures and Tables

**Figure 1 diagnostics-14-00219-f001:**
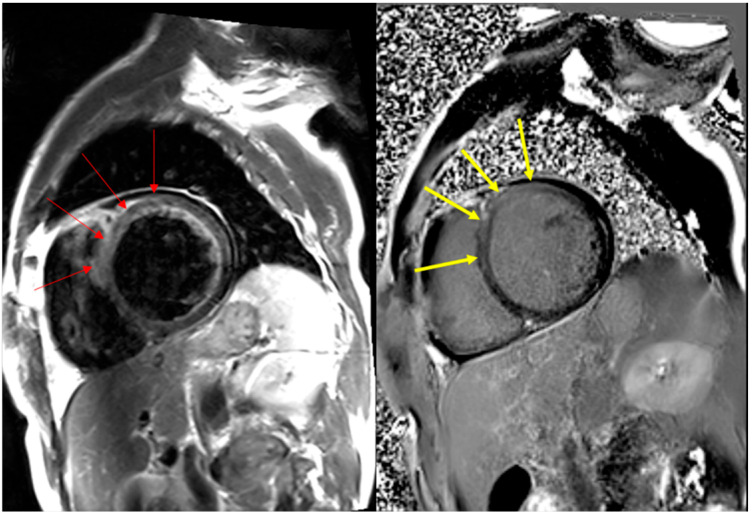
Cardiac magnetic resonance imaging findings of Case 1, post-COVID-19 vaccination myocarditis (PCVM). T2-weighted black blood image (**left**) and delayed enhancement image (**right**) show diffuse increased signal intensity (as compared with skeletal muscles) due to edema and abnormal subepicardial and mid-wall hyper-enhancement, particularly in the anterior and antero-septal portions of the left ventricle (red and yellow arrows). Reprinted in part with modification from Ref. [14].

**Figure 2 diagnostics-14-00219-f002:**
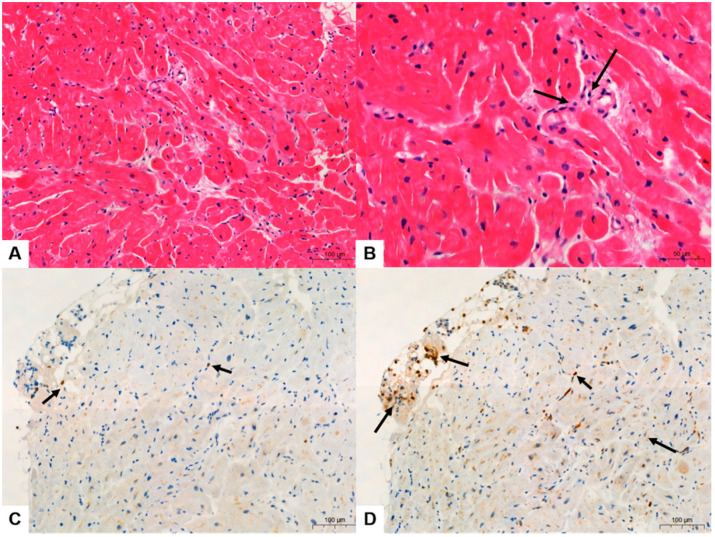
Case 1, post-COVID-19 vaccination myocarditis (PCVM). (**A**,**B**) Endomyocardial biopsy (EMB) showed active multifocal lymphocytic myocarditis (LM) with a few microfoci of myofiber eosinophilic changes and infiltrating lymphocytes (black arrows in (**B**)) (hematoxylin and eosin (H&E), 200× in (**A**) and 400× in (**B**)). (**C**) Infiltrating CD3-positive lymphocytes (black arrows) were 6/mm^2^. (**D**) Infiltrating CD68-positive histiocytes (black arrows) were 10/mm^2^ (IHC, 400× in (**C**,**D**)).

**Figure 3 diagnostics-14-00219-f003:**
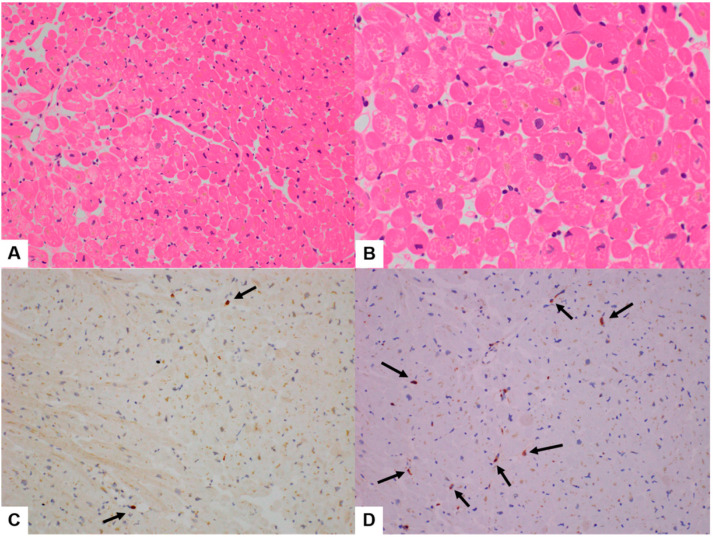
Case 4 borderline post-COVID-19 vaccination myocarditis (PCVM). (**A**,**B**) EMB showed a few lymphocytic infiltrations with a few microfoci of myofiber eosinophilic changes and disrupted myocytes (H&E, 200× in (**A**) and 400× in (**B**)). (**C**) Infiltrating CD3-positive lymphocytes (black arrows) were 2/mm^2^. (**D**) Infiltrating CD68-positive histiocytes (black arrows) were 7/mm^2^ (IHC, 400× in (**C**,**D**)).

**Figure 4 diagnostics-14-00219-f004:**
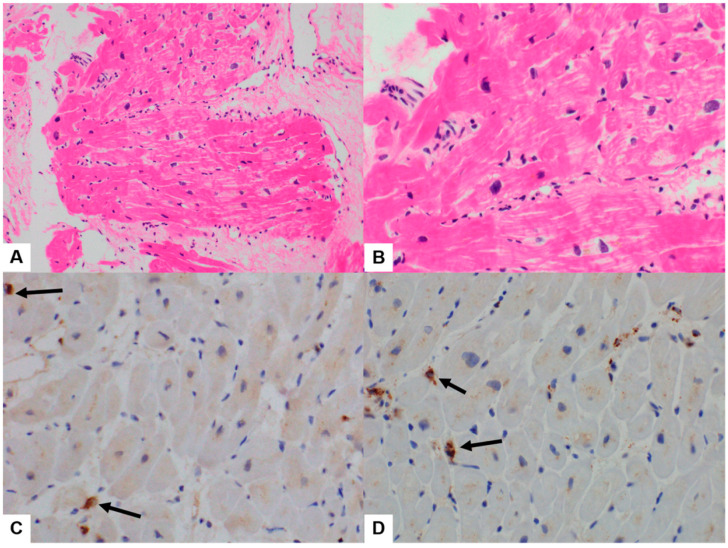
Case 6 of post-COVID-19 vaccination myocarditis (PCVM). (**A**,**B**) EMB showed a few lymphocytic infiltrations with a few microfoci of myofiber eosinophilic changes and disrupted myocytes (H&E, 200× in (**A**) and 400× in (**B**)). (**C**) Infiltrating CD3-positive lymphocytes (black arrows) were 10/mm^2^. (**D**) Infiltrating CD68-positive histiocytes (black arrows) were 5/mm^2^ (IHC, 400× in (**C**,**D**)).

**Table 1 diagnostics-14-00219-t001:** Clinicopathologic findings of previous cases with histopathologic examinations and our six cases of post-COVID vaccination myocarditis.

Case Series [Reference No.]	Age (Yrs)	Sex	Cardiac Function LVEF (%)	Symptoms	MRI Findings	CK-MB	Cardiac Troponin * I or # T	Pathologic Myocarditis Type	Specimen Type	Vaccine Type (Dose)	Symptoms Onset after Vaccination (Days)
Rosner et al. [6]	28	M	51	chest pain, no fever or coughing	Patchy LGE, no ED	U	* 17.08 (ng/mL)	U	ND	J (1st)	5
Rosner et al. [6]	39	M	35–40	chest pain, no fever or coughing	Patchy LGE, no ED	U	* 11.01 (ng/mL)	U	ND	P (2nd)	3
Rosner et al. [6]	39	M	60	chest pain, fever, chills, and shortness of breath	Multifocal LGE, no ED	U	* 13.01 (ng/mL)	U	ND	P (2nd)	4
Rosner et al. [6]	24	M	53	chest pain	Midmyocardial LGE, ED	U	* 0.37 (ng/mL)	U	ND	P (1st)	7
Rosner et al. [6]	19	M	55	chest pain	Multifocal LGE, ED	U	* 44.8 (ng/mL)	U	ND	P (2nd)	2
Rosner et al. [6]	20	M	50–55	chest pain	Subepicardial LGE, ED	U	* 8.36 (ng/mL)	U	ND	P (2nd)	3
Rosner et al. [6]	23	M	58	subjective fevers, chest pain, and myalgia	Midmyocardial LGE, no ED	U	U	U	ND	P (2nd)	3
Abbate et al. [4]	27	M	20	nausea and vomiting	U	252 (ng/ml)	U	U	ND	P (1st)	2
Abbate et al. [4]	34	F	15	fever, cough, chest pain, nausea, and vomiting	U	42.4 (ng/ml)	U	U	ND	P (2nd)	9
Larson et al. [5]	22	M	50	fever, chills, and myalgia	LGE	U	285 FUL	U	ND	M (2nd)	3
Larson et al. [5]	31	M	34	fever, chills, and myalgia	LGE	U	46 FUL	U	ND	M (2nd)	3
Larson et al. [5]	40	M	47	chest pain	ED, LGE, PE	U	520 FUL	U	ND	P (2nd)	2
Larson et al. [5]	56	M	60	chest pain	ED, LGE	U	37 FUL	U	ND	P (2nd)	3
Larson et al. [5]	26	M	60	cough, fever	ED, LGE, PE	U	100 FUL	U	ND	P (2nd)	3
Larson et al. [5]	35	M	50	fever and chest pain	ED, LGE	U	29 FUL	U	ND	P (2nd)	2
Larson et al. [5]	21	M	54	fever and chest pain	ED, LGE, PE	U	1164 FUL	U	ND	P (2nd)	4
Larson et al. [5]	22	M	53	chest pain	ED, LGE	U	1433 FUL	U	ND	M (2nd)	2
Verma et al. [11]	45	F	15–20	dyspnea and dizziness	LGE	U	* 10.453 (ng/mL)	LM admixed with eosinophils, B cells, and plasma cells	EMB	P (1st)	10
Verma et al. [11]	42	M	15	dyspnea and chest pain	Not available	U	* 44.30 (ng/mL)	An inflammatory infiltrate admixed with macrophages, T cells, eosinophils, and B cells	Autopsy	M (2nd)	2wks
Ujueta et al. [10]	62	F	29	progressive body aches, weakness, and fatigue	U	U	* 6.4 (ng/mL)	LM with sparse eosinophils	Autopsy	J (1st)	4
Sung et al. [9]	63	M	35	fever, fatigue, and cough	U	U	# 5.816 (ng/mL)	GCM	EMB	P (2nd)	7
Kiblboek et al. [8]	18	M	33	chest pain, fever, and fatigue	ED, LGE	U	# 1.386 (ng/mL)	LM with sparse eosinophils	EMB	P (1st)	3
Kiblboek et al. [8]	22	M	40	chest pain, no fever, and fatigue	ED, LGE	U	* 38.735 (ng/mL)	LM with sparse eosinophils	EMB	P (2nd)	1
Kiblboek et al. [8]	38	M	48	chest pain, no fever, and fatigue	ED, LGE	U	# 1.104 (ng/mL)	LM with sparse eosinophils	EMB	P (1st)	4
Yamamoto et al. [12]	41	M	15	chest pain, myalgia, and fever	U	236 (U/L)	# 15 (ng/mL)	Severe LM predominantly composed of cytotoxic T cells (CD8+) and macrophages (CD68+) admixed with B cells (CD20+) and a few eosinophils	EMB	M (2nd)	19
Yamamoto et al. [12]	18	M	27	fever, chest pain, and fatigue	U	32 (U/L)	# 3.09 (ng/mL)	Severe LM predominantly composed of cytotoxic T cells (CD8+) and macrophages (CD68+)	EMB	P (1st)	9
Yamamoto et al. [12]	18	M	46	fever and chest pain	U	72 (U/L)	# 1.3 (ng/mL)	Inflammatory cell infiltration is trivial, obvious cardiomyocyte damage (loss of nuclei and mild vacuolar degeneration) and perivascular and interstitial fibrosis	EMB	M (2nd)	2
Yamamoto et al. [12]	18	M	62	fever and chest pain	U	32 (U/L)	# 0.515 (ng/mL)	Same as the above	EMB	M (2nd)	3
Chow et al. [7]	45	F	40	progressive decline of exercise capacity, palpitation, fatigue, and exertional dyspnea	Multifocal LGE	U	U	LM and a focal histiocytic collection suggestive of an ill-defined granuloma	EMB	M (1st)	7
Our case 1	59	M	24	dyspnea, leg edema	Patchy midmyocardial LGE	22.7 (ng/mL)	# 0.019 (ng/mL)	LM	EMB	P (1st)	3
Our case 2	52	M	35	cardiac arrest d/t ventricular fibrillation	ND	31.3 (ng/mL)	# 2.00 (ng/mL)	LM	EMB	P (2nd)	19
Our case 3	19	M	65	chest pain	LGE	11.7 (ng/mL)	# 0.085 (ng/mL)	LM	EMB	M (2nd)	3
Our case 4	83	M	35	dyspnea, palpitation due to ventricular tachycardia	LGE	18.7 (ng/mL)	# 0.82 (ng/mL)	LM	EMB	P (3rd)	2
Our case 5	69	F	60	dyspnea, chest discomfort	Patchy subepicardial and midmyocardial LGE	23.4 (ng/mL)	# 0.667 (ng/mL)	LM	EMB	M (3rd)	2
Our case 6	38	M	18	myalgia, dyspnea	Patchy subepicardial and midmyocardial LGE and PE	15.96 (ng/mL)	# 0.233 (ng/mL)	LM	EMB	P (3rd)	12

LVEF, left ventricular ejection fraction; ND, testing not done; U, unknown; J, Ad26.COV2.S viral vector (Janssen) COVID-19 vaccine (Johnson & Johnson, New Brunswick, NJ, USA); P, BNT162b2 vaccine (Pfizer-BioNTech, New York, NY, USA); M, mRNA-1273 vaccine (Moderna, Cambridge, MA, USA); LM, lymphohistiocytic myocarditis; ED, edema; LGE, late gadolinium enhancement; PE, pericardial effusion; * Troponin I, # Troponin T; FUL: fold of the upper limit; GCM, giant-cell myocarditis; EMB, endomyocardial biopsy.

**Table 2 diagnostics-14-00219-t002:** Quantitative analysis of CD3+ lymphocytes and CD68+ histiocytes of post-COVID vaccination myocarditis.

Case [Reference No.]	Age (Years)	Sex	Myocarditis Type	CD3 (Cells/mm^2^)	CD68 (Cells/mm^2^)
Kiblboeck et al. [8]	18	M	LM with sparse eosinophils	>50	>100
Kiblboeck et al. [8]	22	M	LM with sparse eosinophils	5	14
Kiblboeck et al. [8]	38	M	LM with sparse eosinophils	30	34
Our case 1	59	M	LM (* myocarditis)	6	10
Our case 2	52	M	LM (* borderline myocarditis)	5	5
Our case 3	19	M	LM (* myocarditis)	10	5
Our case 4	83	M	LM (* borderline myocarditis)	2	7
Our case 5	69	F	LM (* borderline myocarditis)	3	0
Our case 6	38	M	LM (* myocarditis)	10	5

LM, lymphohistiocytic myocarditis; * classified according to the Dallas criteria.

## Data Availability

Data are available on reasonable request.

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
