# Peer review of "Post-COVID-19 Vaccination Myocarditis: A Histopathologic Study on a Monocentric Series of Six Cases"

_diagnostics, 2024, doi:10.3390/diagnostics14020219_

Round 1
Reviewer 1 Report
Comments and Suggestions for Authors
The controversial issue of myocarditis occurring after COVID-19 vaccination resides on whether it is an immunological reaction following vaccination or whether a true infection of viral settlement in the myocardium. To clarify the question, the employment of endomyocardial biopsy deserves molecular investigation for a gold standard diagnosis. This is a limitation of the study to be stress at the end of the discussion. Dallas criteria, based on cardiomyocite necrosis, are overcome.
Comments on the Quality of English LanguageMinor editing of English language required
Author Response
Comments and Suggestions for Authors
The controversial issue of myocarditis occurring after COVID-19 vaccination resides on whether it is an immunological reaction following vaccination or whether a true infection of viral settlement in the myocardium. To clarify the question, the employment of endomyocardial biopsy deserves molecular investigation for a gold standard diagnosis. This is a limitation of the study to be stress at the end of the discussion. Dallas criteria, based on cardiomyocite necrosis, are overcome.
Answer: Thank you for your comment. We agree with the reviewer’s opinions, about the controversial issue of myocarditis occurring after COVID-19 vaccination resides on whether it is an immunological reaction following vaccination or whether a true infection of viral settlement in the myocardium. Although we have not done the molecular investigation to identify myocarditis occurring after COVID-19 vaccination in this study, previously we conducted a single cell RNA sequencing and single-cell T cell receptor sequencing analyses of peripheral blood mononuclear cells (PBMCs) using the samples obtained from one of the patients (case no. 1) and published the result. We have revised that finding in the discussion.
Comments on the Quality of English Language
Minor editing of English language required
Answer: We have utilized the professional English editing service in writing and revising this manuscript.
Reviewer 2 Report
Comments and Suggestions for Authors
Dr Ahn and coworkers aimed to describe a series of biopsy-proven myocarditis cases occurring after anti-COVID vaccine administration. They retrospectively collected six cases.
Clarification is required regarding methodology: inclusion criteria are not adequately listed (age range, days from vaccination, time range of data collection, troponin data and/or imaging available); criteria for the diagnosis of myocarditis on endomyocardial biopsies (EMB) are not specified (do the authors apply just the Dallas criteria? Was molecular search for viral genomes performed on EMB?); acquirement of clinical data is severely limited (ECG, troponin, symptoms, cardiac magnetic resonance etc).
Regarding histopathological analysis, the authors report the diagnosis of myocarditis at EMB in all the six cases (supposedly applying only Dallas criteria).
Case 1 is listed as lymphohistiocytic myocarditis, even if the number of lymphocytes is just 6/mm2; the image related to this case didn’t show any lymphocyte cluster and myocytes are normal, despite some lipofuscin deposits; moreover, the lymphocyte count is not performed on the same field as illustrated by H&E, but instead it seems to have been applied on a deposition of thrombotic material.
Case 2 is listed as granulomatous myocarditis, but the highlighted fragment on Fig. 2 is entirely composed of blood material (fibrin and leucocytes), commonly found on EMB specimens and unrelated to diagnosis. In addition, in the discussion session the authors report this patient as having suffered cardiac arrest with hypoxic brain damage. Was EMB performed before or after cardiac arrest?
Case 3, listed as lymphohistiocytic myocarditis, apparently reach the threshold of 7 CD3+ cells/mm2, but the image on Fig. 3 is exactly the same of that in Fig. 1 illustrating case 1 (possible mistake?).
It could be interesting to check images of the remaining cases, possibly to exclude myocarditis also in them (2 or 3 lymphocytes/mm2 are certainly not forming a significant cluster).
Minor comments
· Table 1: reports in which EMB resulted negative but post-vaccination myocarditis was clinically diagnosed should be reported [Larson, Rosner, Abbate].
· Relevant clinical data of the patients should be reported in the Results section and not in the Discussion.
· References are inaccurate (e.g., Verma reported as ref. 8 is not included in the list of references)
· Conclusion is rather confusing: the authors state that 2/6 cases did not fulfill Dallas criteria, so then what criteria did they apply to diagnose myocarditis? Probably they meant that some cases were diagnosed as borderline myocarditis according to Dallas criteria.
· The sentence “Practicing pathologists should be aware of the so-called Dallas criteria when interpreting histopathological findings of EMB samples from PCVM” has no clear significance. Probably every pathologist worldwide analyzing EMB samples is aware of the Dallas criteria; some pathologists apply them, some apply European Society of Cardiology criteria, others use both. The sentence should be rephrased.
References
Abbate A, Gavin J, Madanchi N, et al S. Fulminant myocarditis and systemic hyperinflammation temporally associated with BNT162b2 mRNA COVID-19 vaccination in two patients. Int J Cardiol. 2021 Oct 1;340:119-121. doi: 10.1016/j.ijcard.2021.08.018. Epub 2021 Aug 18. PMID: 34416319; PMCID: PMC8372420.
Larson KF, Ammirati E, Adler ED, et al. Myocarditis After BNT162b2 and mRNA-1273 Vaccination. Circulation. 2021 Aug 10;144(6):506-508. doi: 10.1161/CIRCULATIONAHA.121.055913. Epub 2021 Jun 16. PMID: 34133884; PMCID: PMC8340725.
Rosner CM, Genovese L, Tehrani BN, et al. Myocarditis Temporally Associated With COVID-19 Vaccination. Circulation. 2021 Aug 10;144(6):502-505. doi: 10.1161/CIRCULATIONAHA.121.055891. Epub 2021 Jun 16. PMID: 34133885; PMCID: PMC8340723.
Verma AK, Lavine KJ, Lin CY. Myocarditis after Covid-19 mRNA Vaccination. N Engl J Med. 2021 Sep 30;385(14):1332-1334. doi: 10.1056/NEJMc2109975. Epub 2021 Aug 18. PMID: 34407340; PMCID: PMC8385564.
Comments on the Quality of English LanguagePlease see general comments.
Author Response
Comments and Suggestions for Authors
Dr Ahn and coworkers aimed to describe a series of biopsy-proven myocarditis cases occurring after anti-COVID vaccine administration. They retrospectively collected six cases.
Clarification is required regarding methodology: inclusion criteria are not adequately listed (age range, days from vaccination, time range of data collection, troponin data and/or imaging available); criteria for the diagnosis of myocarditis on endomyocardial biopsies (EMB) are not specified (do the authors apply just the Dallas criteria? Was molecular search for viral genomes performed on EMB?); acquirement of clinical data is severely limited (ECG, troponin, symptoms, cardiac magnetic resonance etc).
Answer: We have applied Dallas criteria for the diagnosis of myocarditis on endomyocardial biopsies (EMB) in this study and specified. We did not perform the molecular search for viral genomes with the EMB samples in this study. Instead, we have reviewed the medical records and excluded that the possibility of other causes, such as viral or autoimmune diseases, serologically: negative serological marker tests for other viruses, such as adenovirus, coxsackievirus B1 and parvovirus. And we have checked that all the patients were negative for the auto-immune disease markers. We have provided more clinical data (ECG, troponin, symptoms, and cardiac magnetic resonance) in Table 1.
Regarding histopathological analysis, the authors report the diagnosis of myocarditis at EMB in all the six cases (supposedly applying only Dallas criteria).
Case 1 is listed as lymphohistiocytic myocarditis, even if the number of lymphocytes is just 6/mm2; the image related to this case didn’t show any lymphocyte cluster and myocytes are normal, despite some lipofuscin deposits; moreover, the lymphocyte count is not performed on the same field as illustrated by H&E, but instead it seems to have been applied on a deposition of thrombotic material.
Answer: We have changed Figures 1B—D, in which disrupted myocytes are more prominent and CD3+ lymphocytes and CD68+ histiocytes are prominent and depicted and renumbered Figure 1 as Figure 2.
Case 2 is listed as granulomatous myocarditis, but the highlighted fragment on Fig. 2 is entirely composed of blood material (fibrin and leucocytes), commonly found on EMB specimens and unrelated to diagnosis. In addition, in the discussion session the authors report this patient as having suffered cardiac arrest with hypoxic brain damage. Was EMB performed before or after cardiac arrest?
Answer: We have comprehended your concern, however, we interpreted these aggregated CD68+ cells are histiocytes (monocytes), because on the HE stain, these cells showed large, eccentrically placed nuclei which stains less intensely with more open chromatin than other leucocytes. Additionally, characteristic nuclear indentations are denoted in the nuclei. Then, we changed Figure 2B of high magnification to highlight these cytomorphologic features, and renumbered Figure 2 as Figure 3. EMB was performed after cardiac arrest on the patient (case 2).
Case 3, listed as lymphohistiocytic myocarditis, apparently reach the threshold of 7 CD3+ cells/mm2, but the image on Fig. 3 is the same of that in Fig. 1 illustrating case 1 (possible mistake?).
It could be interesting to check images of the remaining cases, possibly to exclude myocarditis also in them (2 or 3 lymphocytes/mm2 are certainly not forming a significant cluster).
Answer: We have renumbered Figure 3 as Figure 4. As you recommended, we added Figure 5 showing the pathologic features of borderline myocarditis.
Minor comments
- Table 1: reports in which EMB resulted negative but post-vaccination myocarditis was clinically diagnosed should be reported [Larson, Rosner, Abbate].
Answer: We incorporated these clinically diagnosed cases in Table 1, as you commented.
- Relevant clinical data of the patients should be reported in the Results section and not in the Discussion.
Answer: We have moved the relevant clinical data into the results section.
- References are inaccurate (e.g., Verma reported as ref. 8 is not included in the list of references)
Answer: We have corrected the inaccurate references.
- Conclusion is rather confusing: the authors state that 2/6 cases did not fulfill Dallas criteria, so then what criteria did they apply to diagnose myocarditis? Probably they meant that some cases were diagnosed as borderline myocarditis according to Dallas criteria.
Answer: We have changed conclusion as you mentioned.
- The sentence “Practicing pathologists should be aware of the so-called Dallas criteria when interpreting histopathological findings of EMB samples from PCVM” has no clear significance. Probably every pathologist worldwide analyzing EMB samples is aware of the Dallas criteria; some pathologists apply them, some apply European Society of Cardiology criteria, others use both. The sentence should be rephrased.
Answer: We have rephrased that sentence.
References
Abbate A, Gavin J, Madanchi N, et al S. Fulminant myocarditis and systemic hyperinflammation temporally associated with BNT162b2 mRNA COVID-19 vaccination in two patients. Int J Cardiol. 2021 Oct 1;340:119-121. doi: 10.1016/j.ijcard.2021.08.018. Epub 2021 Aug 18. PMID: 34416319; PMCID: PMC8372420.
Larson KF, Ammirati E, Adler ED, et al. Myocarditis After BNT162b2 and mRNA-1273 Vaccination. Circulation. 2021 Aug 10;144(6):506-508. doi: 10.1161/CIRCULATIONAHA.121.055913. Epub 2021 Jun 16. PMID: 34133884; PMCID: PMC8340725.
Rosner CM, Genovese L, Tehrani BN, et al. Myocarditis Temporally Associated With COVID-19 Vaccination. Circulation. 2021 Aug 10;144(6):502-505. doi: 10.1161/CIRCULATIONAHA.121.055891. Epub 2021 Jun 16. PMID: 34133885; PMCID: PMC8340723.
Verma AK, Lavine KJ, Lin CY. Myocarditis after Covid-19 mRNA Vaccination. N Engl J Med. 2021 Sep 30;385(14):1332-1334. doi: 10.1056/NEJMc2109975. Epub 2021 Aug 18. PMID: 34407340; PMCID: PMC8385564.
Reviewer 3 Report
Comments and Suggestions for Authors
Dear Authors,
Congratulation Yours study.
Did the described patients have only myocarditis or also pericarditis ?
How were the described patients treated ? Antiviral ?
Was LGE (late gadolinium enhancement) observed in the MRI performed ?
Best regards
Author Response
Comments and Suggestions for Authors
Dear Authors,
Congratulation Yours study.
Answer: Thank you for your positive feedback about our study. Your valuable comment is helpful in enhancing the value of our research. We have revised and addressed all the comments.
Did the described patients have only myocarditis or also pericarditis ?
Answer: In this study, the described patients have only myocarditis and no patient does have pericarditis. Symptoms of two patients were mild, with a normal range of LVEF, and had no sequelae (Case no. 3, LVEF 65%; Case no. 5, 60%; Table 1).
How were the described patients treated ? Antiviral ?
Answer: The Symptoms of two patients were mild, with a normal range of left ventricular function, and had no sequelae. Two patients with mild symptoms received conservative therapy, whereas the remaining four received treatment for heart failure. No immunosuppressive treatments were administered.
Was LGE (late gadolinium enhancement) observed in the MRI performed ?
Answer: LGE (late gadolinium enhancement) was observed in all the performed MRI on our patients. We have summarized cardiac MRI findings in Table 1.
Round 2
Reviewer 2 Report
Comments and Suggestions for Authors
Main issues were not resolved: Fig. 2 doesn’t represent a myocarditis case (no cluster of lymphocytes is evident), Fig. 3 and 4 just illustrate thrombotic material, Fig. 5 includes only normal myocardium. These are not biopsy-proven myocarditis cases, according to internationally-approved pathological criteria.
Author Response
Answer: We agreed with the reviewer that our cases are not biopsy-proven myocarditis cases. The gold standard for diagnosing myocarditis is endomyocardial biopsy (EMB). However, for some reasons such as risk of complications, performing of EMB is very limited in myocarditis in the practice. As our patients had clinical features that could be considered myocarditis, such as late gadolinium enhancement in heart MRI, increased cardiac marker and CRP, and symptoms like chest pain and dyspnea, they have been clinically diagnosed of myocarditis. In this series, two cases met the > 7 CD3-positive T lymphocytes/mm2 Dallas criterion (case nos. 3 and 6), but the remaining four did not. However, three cases met the ≥14 leukocytes in the myocardium criterion, including up to 4 monocytes/mm2 (Case nos. 1, 2, 3, and 6), whereas the remaining three did not. These (Case nos. 2, 4 and 5) could be categorized as “borderline” myocarditis and repeat biopsy may be indicated (Table 2). So, we commented in the abstract and main text that most PCVM cases showed mild degree inflammation histopathologically and some cases could not fulfill the Dallas criteria and classified as borderline myocarditis.
We have changed Fig. 2 as a more representative microphotograph of the case 1. We have changed Fig. 3 as a microphotograph of the case 4 and changed Fig. 4 as a microphotograph of case 6. We agreed that Fig. 3 and 4 demonstrate thrombotic material, therefore we recounted CD68+ cells of both case 2 and 3 and reclassified our case 2 as lymphohistiocytic myocarditis (LM) (Table 2). We have deleted Fig. 5.